# How Should Cryptocurrencies Be Defined and Reported? An Exploratory Study of Accounting Professor Opinions

D. Larry Crumbley [1], Donald L. Ariail [2,*] and Amine Khayati [2]

1   Department of Accounting and Finance, Texas A&M University—Corpus Christi, RELLIS Campus, 3100 TX-47, Bryan, TX 77807, USA; donald.crumbley@tamucc.edu
2   Coles College of Business, Kennesaw State University, Kennesaw, GA 30144, USA; akhayati@kennesaw.edu
*   Correspondence: dariail1@kennesaw.edu

**Abstract:** Crypto assets have upset the pillars of regulatory and centralized monetary policy, and the Financial Accounting Standards Board (FASB) has been slow in developing a position on how to account for cryptocurrencies. Currently, there are many accounting, finance, and tax meanings of cryptocurrencies. The purpose of this study is to show the path FASB has taken to develop accounting standards for more than 20,000 crypto assets, outline the positions other authorities and agencies have taken, and discuss Central Bank Digital Currencies since the United States and other countries are considering replacing their fiat currency with a digital currency. Furthermore, the study presents insights from an exploratory survey of accounting faculty opinions on cryptocurrencies. The discussion of virtual currency regulatory and accounting treatments informs the development of a regulatory framework.

**Keywords:** accounting regulation; cryptocurrency; FASB

## 1. Introduction

The onset of cryptocurrency has shaken the pillars of regulated and centralized monetary policy. Cryptocurrencies (crypto) or altcoins are increasingly being used by individuals and companies for their online transactions and financial investments (Ramassa and Leoni 2022). With the gradual acceptance of cryptocurrency, the need to define a regulatory and accounting framework is urgently needed. Currently, there are many accounting, finance, and tax meanings of crypto assets. This exploratory study contributes to the cryptocurrency dialog by assessing the opinions of accounting faculty regarding how this virtual asset should be defined and reported in financial statements.

Specifically, the survey solicited input on 26 virtual-currency-related questions. Responses, which were analyzed using descriptive statistics, were grouped into three categories: regulatory treatment, accounting treatment, and cryptocurrency impact. The purpose of this study is to gain insights regarding accounting faculty perceptions of how crypto assets should be defined and reported. Potential options included classifying digital assets as either tangible or intangible, as currency, as cash or cash equivalent, as a commodity, and as a financial instrument. The classification of crypto assets may impact financial and tax reporting and financial disclosure. Examples of this impact include the reporting of value and the recognition of impairment and gain. Furthermore, virtual currency perceptions of accounting faculty can inform regulators regarding how digital policies may impact issues such as tax fraud, individual privacy, and banking systems. The present study's exploratory findings contribute to the emerging literature on virtual currency regulation and taxonomy, provide guidance to regulators, and present a rich landscape of information that will be of interest to future researchers.

## 2. Literature Review

Ankenbrand et al. (2020) proposed a universal approach to the classification of all assets, including digital and cryptographic assets. Supported by the literature, their taxonomy includes 14 distinct asset attributes: claim structure, technology, underlying, consensus-validation mechanism, legal status, governance, information complexity, legal structure, information interface, total supply, issuance, redemption, transferability, and fungibility. They concluded that their asset taxonomy helps to bridge the gap between physical and digital assets.

Di Matteo et al. (2021) developed a taxonomy for investigating and categorizing papers that discussed initial coin offerings (IOC). According to Investopedia (2023) "an initial coin offering is cryptocurrency's first issue to the public. Often accompanied by a white paper describing the technology behind it, the investing audience determines its value". Based on their review of 226 papers published from 2017 through 2021, Di Matteo et al.'s (2021) taxonomy included seven dimensions: research approach, research design, data collection, philosophical view, focus, research issue, and ICO phase.

Watters (2023) used doctrinal analysis of case law, statues, and jurisprudence from multiple jurisdictions to explore the nuances of three broad categories of digital assets termed decentralized digital assets. The three categories were commodities (stores of value), currencies (mediums of exchange), and securities (held with the expectation of profit). The author concluded that the ambiguity of decentralized digital asset terminology along with inconsistently applied standards calls for the development of "...a simplified framework ... [based on] a rebuttable presumption that the sale of an asset is the for the asset's intended use case" (p. 363).

Fokri et al. (2021) discussed five classifications of digital assets: coins, currency, tokens, payment tokens, utility tokens, and security or asset tokens. The author called for Islamic scholars to develop consistent classification guidelines in accord with sharia law for blockchain technology. Specific guidelines for this technology are needed since "...if the whole world uses blockchain technology and digital currency as a medium of economic activities, then, countries that do not use that technology face difficulties when collaborating or cooperating with other countries" (p. 1358).

Taken together, these recent studies highlight the complexity of defining digital assets and the challenges involved in developing a consistent standardized taxonomy framework. Unclear and varying standards among jurisdictions create ambiguity and higher compliance costs for accounting treatment and reporting.

## 3. FASB's Path to Crypto Standards

Global ownership of crypto assets is estimated at 420 million owners, with 54 million in the U.S. (Triple 2023). Coingecko.com lists 23 companies in July 2023 that are buying Bitcoins as part of their treasury. The top three companies doing so are MicroStrategy, Inc, Marathon Digital, and Coinbase. Tesla is not on the list, but Telsa lost USD 140 million in 2022 alone on its Bitcoin sales. Luo and Yu (2022) indicated that 40 companies own Bitcoins. Although Bitcoins are illegal or restricted by some countries (e.g., China, Iran, Russia), they are permitted in around 111 countries (Crypto News 2023). However, cryptos are a cost-effective financial instrument for some countries that are dependent on remittance income (Oxford Analytica 2023).

In their 2023 Crypto Crime Report, Chainalysis indicated that in 2022 "...illicit crypto transaction volume rose for the second consecutive year, hitting an all-time high of $20.6 billion" (Chainalysis Team 2023, p. 5). Crypto scam revenue dropped (perhaps due to a drop in crypto prices) by 46% from 2021 to 2022 (from USD 10.9 billion in 2021 to USD 5.9 billion in 2022). On the other hand, money laundering, identified as cryptocurrency amounts sent to illicit addresses, grew by 68%: from USD 14.2 billion in 2021 to USD 23.8 billion in 2022. Nevertheless, "...43% of 2022's illicit transaction volume came from activity associated with sanctioned entities" (Chainalysis Team 2023, p. 5), such as Garantex, Hydra Marketplace, and Tornado Cash.

Although the U.S. ranks fifth on the Chainalysis Global Crypto Adoption Index behind Vietnam, the Philippines, Ukraine, and India, the U. S. Financial Accounting Standards Board (FASB) has been slow to develop a position on how to account for cryptocurrencies. This Index ranks 146 countries, with emerging markets dominating it, and the global adoption of cryptos levels off in 2022 during its bear market (Chainalysis Team 2022).

However, finally, in December 2019, the American Institute of Certified Public Accountants (AICPA Digital Asset Work Group 2019) and the Chartered Institute of Management Accountants (CIMA) presented nonauthoritative guidance from a Digital Asset Working Group, which was updated in June 2022, and the 47-page Practice Aid made a number of suggestions (AICPA Digital Asset Work Group 2019):

- Cryptocurrencies or virtual assets should be treated as indefinite-lived intangible assets under Topic 350.
- Cryptocurrencies should be set up at cost and tested for impairment at least annually since the indefinite life is not subject to amortization (assuming for investment or treasury).
- If impaired, impairment losses should be recognized as an expense in the income statement and should not be marked up if they recover.
- When impaired, under U.S. GAAP, this should be written downward in value, with an expense in the income statement.

After a great deal of internal activities and discussions, on 23 March 2023, FASB issued a Proposed Accounting Standards Update for Intangibles—Goodwill and Other Crypto Assets (Subtopic 350-60), with a comment deadline of 6 June 2023. This new proposed accounting treatment applies to crypto assets that meet the following six criteria (FASB 2023):

1. Meet the definition of intangible assets as defined in the Codification Master Glossary. In other words, the assets must lack physical substance but not be a financial asset.
2. Do not provide a holder with enforceable rights to, or claims on, underlying goods, services, or other assets.
3. Created or reside on a distributed ledger based on blockchain technology.
4. Secured through cryptography.
5. Fungible.
6. Not created or issued by the reporting entity or its related parties.

The proposed standards were adopted on 6 September 2023. Public and private entities must eventually use Topic 820 Fair Value Measurement to determine the initial fair value of the more than 20,000 cryptocurrencies. The new standards become effective after 15 December 2024 for fiscal year companies, and early adoption is allowed. However, companies must currently use ASC 350 and treat crypto assets as indefinite-lived intangible assets. Fair value is the price that would be received to sell an asset or paid to transfer a liability in an orderly transaction between market participants at the measurement date. Furthermore, increases and decreases in fair value should be recognized in the comprehensive income for each reporting period. Costs such as commissions should be expensed unless specialized industry measurement guidance requires otherwise (FASB 2023). Note that any capital gain income of a corporation from crypto transactions is normally treated as ordinary income for tax purposes, and long-term and short-term losses are deductible against capital gains with excess losses carried forward to future years.

While the standards are in the proposal stage, if crypto assets are received as noncash consideration in the ordinary course of business (e.g., in exchange for the transfer of goods and services to a customer) and are converted almost immediately into cash, the entity is required to classify these cash receipts as cash flows from operating activities (FASB 2023). For tax years when the new standards are not adopted by a company, there are no specific standards except being classified as indefinite-lived intangible assets similar to copyrights. Thus, crypto is set up at cost and reviewed for impairment purposes. If written down, no future upward adjustments are available.

Moreover, an entity must present (1) crypto assets measured at fair value separately from other intangible assets in the balance sheet and present (2) changes in the fair value measurement of crypto assets separately from changes in the carrying amounts of other intangible assets in the income statement (or statement of changes in net assets for not-for-profit organizations). Once an entity adopts the fair value method, there will be a need for a cumulative-effect adjustment to the opening balance of retained earnings (FASB 2023).

Apparently, FASB will not provide guidance on determining the fair value of virtual currency (VC) assets, but the IRS accepts the value as determined by a cryptocurrency explorer or blockchain explorer (IRS n.d.). Until FASB issues final accounting standards, cryptos should be recorded at historical cost and must be periodically tested for impairment. Any impairment charge is to be shown as an expense, and any future increases in value are to be ignored. FASB has not taken a position on how a governmental virtual currency would be treated, but it seems a government virtual currency would be treated as legal tender rather than an intangible asset. In addition, non-fungible tokens (NFTs) are not addressed, and financial statement presentation and required disclosures are not covered. Furthermore, there is no mention of Central Bank Virtual Currencies (CBVC).

*Other Conflicting Positions Taken by Authorities*

The borderless nature of virtual currency requires countries to pass legislation and to cooperate. Effective policies for crypto assets have become a key policy priority for authorities (IMF Policy Paper 2023). On 10 October 2019, former Chairperson Heath P. Tarbert, of the U.S. Commodity Futures Trading Commission (CFTC) stated that Bitcoins and Ether are commodities and will be regulated by the Commodity Exchange ACT (CEA) of 1936. He also said that similar assets will be treated similarly (Tarbert 2019). The business of the CFTC is to promote integrity, resilience, and vibrancy in the U.S. derivative markets through sound regulations.

CFTC Bitcoin Basics (n.d.) states that Bitcoin is a convertible virtual currency and is determined to be a commodity under the CEA. Furthermore, CFTC "is implicated when a virtual currency is used in a derivative contract, or if there is fraud or manipulation involving a virtual currency traded in interstate commerce" (p. 1). For example, on 6 June 2023, CFTC charged two individuals in Florida for engaging in a deceptive and fraudulent multi-million-dollar Bitcoin fraud (Release Number 8743-23 2023).

The Office of the Comptroller of the Currency's (OCC) mission is to "ensure that national banks and federal savings associations operate in a safe and sound manner, provide fair access to financial services, treat customers fairly, and comply with applicable laws and regulations" (OCC n.d.). Although OCC does not define cryptocurrencies, they issued three Interpretive Letters in 2020 and 2021. Interpretive Letter No. 1179 (2021) clarifies that a bank can legally engage in certain cryptocurrency, distributed ledger, and stable coin activities if the bank can demonstrate to the satisfaction of its supervisory office that there are controls in place to conduct the activity in a safe and sound manner. A stable coin is generally backed by a fiat currency. The bank must notify their supervisory office in writing and wait for approval.

As of March 2023, no crypto asset entity is registered with the SEC or a National Securities Exchange, and no existing national securities exchange currently trades crypto assets. Thus, investors in crypto asset securities may not benefit from laws that protect fraud, manipulations, and other misconduct (SEC 2023). The SEC considers many cryptocurrency coins and tokens to be securities under a so-called Supreme Court Howey test (SEC v. W. H. Howey Co. 1946) of (1) investment of money, (2) in a common enterprise, and (3) with a reasonable expectation of profits to be derived from the efforts of others. Thus, crypto assets could be a commodity or a security, and the SEC has been filing enforcement actions against crypto lenders, exchanges, and crypto companies (Barton et al. 2022). For example, in March 2023, both the DOJ and SEC launched an investigation against SVB Financial Group.

SEC's key attack was against Ripple Labs (SEC v. Ripple Labs, Inc. 2023) for selling XRP tokens in an unregistered security offering, causing confusion with defining cryptocurrency

assets. After two years, the parties decided to skip the trial and allowed the judge to rule over the dispute. Federal Judge Analisa Torres, on 13 July 2023, held that when the XRP tokens are sold to the public, the sale is not considered to be a security. However, when the tokens are sold initially to institutional investors, the sale is an unregistered securities offering. Depending upon other SEC litigation, this summary judgement order could be a pyrrhic victory for the SEC.

The U.S. Department of the Treasury Financial Crimes Enforcement Network (FinCEN) defines cryptocurrency as a medium of exchange that can operate as currency but does have all the attributes of real currency, including legal tender status. FinCEN further defines a convertible virtual currency (CVC) as one that has an equivalent value as real currency or acts as a substitute for real currency. This CVC category includes most cryptocurrencies except Central Bank Digital Currencies such as the Chinese digital yuan (INNREG n.d.).

FinCEN defines an administrator or exchanger of virtual currency as a money service business (MSB), who must register with FinCEN and follow recordkeeping, reporting, and anti-money laundering rules (FIN-2013-G001 2013).

The Internal Revenue Service (IRS) defines cryptocurrencies as a virtual currency (VC) and classifies VCs as property and not currency in Notice 2014-21, 2014-16 IRB 938 2014 (2014). Generally, a person realizes a capital gain or loss on the sale or exchange of cryptocurrencies. Depending upon how the virtual asset is held, the VC could be treated as personal property, investment property, or business property. Davenport and Usrey (2023), however, suggest that the crypto classifications for U.S. tax purposes need to be updated.

Since IFRS has been adopted by 168 jurisdictions (IFRS n.d.), one would believe they would have the final and best procedures for accounting for crypto assets. The IFRS Interpretation Committee (IFRS IC Staff Paper 2019) indicates that VC is not a financial asset but an intangible asset in the scope of IAS 38 *Intangible Assets* or IAS:2 *Inventories* (if held for sale). This position is mandatory for IFRS adopters (Ramassa and Leoni 2022). IAS 38 allows intangible assets to be valued at a cost or a revaluation method. Under the cost model, the asset is valued initially at cost less amortization and impairments. Under the revaluation model, the intangible asset can be revalued where there is an active market.

The Australian Accounting Standard Board appears to suggest use of a VC as inventory should be valued at the lower of cost or realizable value (Venter 2016). Writing in the *Australian Accounting Review*, Tan and Low (2017) suggested that Bitcoins are not intended to be a reporting currency but a complement for fiat money. They suggest that crypto assets could be a financial asset.

The International Monetary Fund (IMF) was initially negative to cryptocurrencies, warning countries against adopting Bitcoins and other cryptocurrencies at any level. However, Bitcoin activities in the countries of El Salvador, Georgia, and Nigeria have moved IMF from its anti-crypto position. For example, El Salvador and the Central African Republic have officially adopted Bitcoins as their official currency (Bastardo 2023). St. Kitts and Nevis are considering switching over to Bitcoin Cash, and Argentina has a Bitcoin futures contract. The IMF now states that effective regulation of crypto assets is necessary (Cuervo et al. 2019).

Cryptocurrency is now illegal according to the People's Bank of China, except for their Central Bank Digital Currency RMB. After 23 September 2021, China has officially prohibited trading, token issuance, and derivatives of crypto assets as well as overseas crypto exchange services. Yan et al. (2022) found that crypto assets are treated as investments (45%), intangible assets (36%), and inventories (19%). A total of 84% of their respondents stated that the value of crypto assets should be reported at fair value, which is different to the IFRSIC's position.

Japan's accounting treatment is based on Practical Solution No. 38 issued in 2018, which treats crypto assets as trading (investment) assets. For information on how crypto assets is presented in financial statements in Japan, see Yanagida (2023).

This paper does not cover the following other stakeholders:

- Financial Stability Board;

- Financial Action Task Force;
- Basel Committee on Banking Supervision;
- International Organization of Securities Commissions;
- Committee on Payments and Infrastructures;
- Organization for Economic Co-operation and Development.

However, a person can find the legal status of cryptocurrencies and four categories of regulations for 45 countries on the Atlantic Council's Cryptocurrency Regulation Tracker (n.d.).

## 4. Central Bank Digital Currencies

At least 93 percent of central banks are researching the implementation of Central Bank Digital Currencies according to a survey by the Bank of International Settlements (Jung 2023). Central Bank Digital Tracker (2023) (CBDT) indicates that 11 countries have fully launched a Central Bank Digital Currency. On their map, the launched countries are the Bahamas, Jamaica, Eastern Caribbean (8 countries), and Nigeria. There are 18 pilot countries including China, which reaches 260 million people, and it is being tested in over 200 scenarios in 2023. A total of 130 countries (98 percent of global GDP) are exploring a CBDC. Moreover, two countries have adopted Bitcoins as a national currency (El Salvador (21 September 2021), Central African Republic), but the decline in cryptos in 2022 has hindered the outcome. In El Salvador, only about 20 percent of the population is using Bitcoins, and there was a great deal of fraud in the initial USD 30 million free launching of their Bitcoins (Sigalos 2022).

In the U.S., a retail CBDC has stalled, but the U.S. is moving forward with a wholesale bank-to-bank CBDC (Central Bank Digital Tracker 2023). On 9 March 2022, President Biden signed Executive Order 14067 2022 (2022) (Ensuring Responsible Development of Digital Assets) aimed at developing a digital asset policy and organizing federal regulators' efforts in this area. The New York Federal Reserve Bank of New York engaged in a wholesale CBDC experiment, called Project Cedar, which shifted the U.S. from research to the development stage (Project Cedar 1 2022).

The New York Federal Reserve's Project Cedar would convert the U.S. dollar into a new virtual currency. They issued a 12-page "Phase One Report" in the fall of 2022 about their research effort to develop a technical wCBDC that could be used by financial institutions to effect wholesale transactions (e.g., interbank payment clearing and settlement). This Phase 1 experiment examined the potential application of distributed ledger technology, such as blockchain (Project Cedar 1). Project Cedar Phase 2 validated their hypothesis "related to interoperability, speed, and atomic settlement, targeting known problem areas within multi-currency wholesale cross-border payments today" (Project Cedar 2, Phase II x Ubin+ 2022, p. 5).

The U.S. Securities and Exchange Commission Investor Advisory Board sent a letter to Gary Gensler stating that "virtually all, if not all, crypto tokens are securities and they, as well as the platforms trading them should comply with the registration, disclosure, anti-fraud provisions and other investor protector provisions of the federal securities laws" (Investor Advisory Committee Letter 2023, p. 2). The 23-person committee encouraged the SEC to be aggressive in bringing enforcement actions against these companies.

In June 2023, the PCAOB issued a 13-page inspection observation about auditing for companies holding cryptocurrencies, but much of the spotlight was about fraud and encouraging external auditors to locate and report crypto assets. The report suggested the following about the presentation of financial statements, including disclosures (PCAOB 2023, p. 13):

> The auditor must evaluate whether the financial statements are presented fairly, in all material respects, in conformity with the applicable financial reporting framework. The auditor will need to understand the public company or broker-dealer's accounting policies regarding crypto assets and related activities, including the related disclosures, and evaluate whether they are consistent with the applicable financial reporting framework. Each crypto asset may have different character-

istics that would need to be considered by the auditor in their evaluation of the public company or broker-dealer's accounting policies.

Little was provided about how to treat cryptocurrencies or show the information on financial statements or in footnotes. Thus, how would a digital U.S. dollar be treated? The data provided by the present survey provide input, including suggestions, from accountants in academia.

## 5. Research Methodology

### 5.1. Survey

A two-part survey was composed of eight demographic questions and 26 virtual-currency-related questions, which were staggered to reduce response biases. The virtual currency questions are included in Table 2. At the end of the survey, respondents were asked to provide additional feedback. Selected examples of these remarks are included in Appendix A.

The survey was administered in a digital format through Qualtrics. The survey questions were preceded by a cover letter detailing the survey's objectives, disclosures, and instructions for entering a sweepstake draw. To maintain the respondents' confidentiality, the survey invited participants to enter the survey's sweepstake by providing contact information in a separate Qualtrics link.

### 5.2. Data Collection

The study relied on accounting professor contact information contained in Hasselback's *Accounting Directory* (Hasselback 2018), which is the only available comprehensive source of information on accounting professors. This issue of Hasselback's, which was the last one published, has previously been successfully used by the present authors and by others (e.g., Bergner et al. 2022; Burton et al. 2023). From this hard copy directory, names and email addresses were digitally compiled. Using Qualtrics, the survey was sent to accounting professors residing in and outside the United States (U.S.). Nevertheless, the majority of Hasselback-Directory-listed accounting professors teach at U.S. institutions. To increase responses, three waves of emails were utilized. After the third email wave, 20 sweepstake winners were randomly selected and electronically forwarded a USD 30 USD gift card. From an initial list of 9039 faculty, 810 emails were undeliverable (recipients had retired or changed jobs), and 255 completed surveys were returned and useable, for a response rate of about 3%.

## 6. Results

### 6.1. Descriptive Statistics

The first part of the survey collected accounting faculty (hereafter, faculty) demographics. Table 1 presents the distribution of faculty demographics. This sample of faculty were mostly over the age of 50 with 65.1% of the faculty being between the ages of 50 and 69. The age distribution was in line with the findings for faculty level, experience, and tenure. Most (80%) of the faculty were tenured, and most were at the associate or professor levels with proportions of 34.9% and 40.4%, respectively. These accounting professors were experienced (80.3% percent of faculty had more than 20 years of academic experience) and held terminal degrees (84.7% held a terminal degree in their field with either a doctorate or a PhD). In addition, 67.1% of faculty were affiliated with universities with teaching and research focuses, 13.7% were affiliated with top-tier research-focused universities, and 19.2% were affiliated with teaching-focused institutions.

**Table 1.** Respondents' demographics.

| Variable | (N = 255) | | Variable | (N = 255) | |
|---|---|---|---|---|---|
| | No. | % | | No. | % |
| Age | | | Tenure | | |
| Under 30 | 1 | 0.4 | No | 51 | 20.0 |
| 30–39 | 22 | 8.6 | Yes | 204 | 80.0 |
| 40–49 | 37 | 14.5 | Education | | |
| 50–59 | 76 | 29.8 | Bachelor | 3 | 1.2 |
| 60–69 | 90 | 35.3 | Masters | 36 | 14.1 |
| 70–79 | 27 | 10.6 | Doctorate/PhD | 216 | 84.7 |
| 80≤ | 2 | 0.8 | University Focus | | |
| Gender | | | Teaching | 49 | 19.2 |
| Male | 158 | 62.0 | Teaching and Research | 171 | 67.1 |
| Female | 97 | 38.0 | Research | 35 | 13.7 |
| Faculty Level | | | Country | | |
| Lecturer | 25 | 9.8 | Australia | 1 | 0.4 |
| Adjunct Professor | 1 | 0.4 | Canada | 14 | 5.5 |
| Assistant Professor | 25 | 9.8 | Denmark | 1 | 0.4 |
| Associate Professor | 89 | 34.9 | Italy | 1 | 0.4 |
| Professor | 103 | 40.4 | Netherlands | 1 | 0.4 |
| Department Chair | 9 | 3.5 | New Zealand | 6 | 2.4 |
| Dean | 3 | 1.2 | South Africa | 1 | 0.4 |
| Experience | | | United Kingdom | 3 | 1.2 |
| Under 10 | 6 | 2.4 | USA | 227 | 89.0 |
| 10–19 | 44 | 17.3 | | | |
| 20–29 | 48 | 18.8 | | | |
| 30–39 | 86 | 33.7 | | | |
| 40–49 | 60 | 23.5 | | | |
| 50≤ | 11 | 4.3 | | | |

This table presents the distribution of accounting faculty's demographics across age, gender, faculty level, experience, tenure, education, university focus and by country.

As previously mentioned, the study solicited faculty included in the Hasselback Accounting Directory list, which predominantly contained faculty from U.S.-based higher education (HE) institutions. As expected, the majority of faculty taught at HE institutions in the U.S. (89%) followed by Canada (5.5%), New Zealand (2.4%), and the United Kingdom (1.2%). Responses were also received from faculty in Australia, Denmark, Italy, the Netherlands, and South Africa.

*6.2. Faculty Opinions*

The 5-Likert scale responses were coded from 1 to 5, with 1 representing "Strongly disagree" up to 5 which is "Strongly agree". Thus, the mean response for each question could take values from 1 to 5. Table 2 presents the distribution of the responses, and the mean and standard deviations of the coded responses for each question. To draw insights from the faculty agreement or disagreement with the survey questions (statements), we tested whether the mean question responses were significantly different from the neutral value of 3. Thus, the statistical significance of the difference between the mean value and the neutral value of 3 indicated aggregate divergences from the neutral opinion. As previously indicated, we concurrently examined survey question (Q) responses related to the topics of regulatory treatment, accounting treatment, and cryptocurrency impact.

**Table 2.** Survey questions' descriptive statistics.

| Questions | Mean | Std. Deviation | Percentage Frequency | | | | | Test of Mean | |
|---|---|---|---|---|---|---|---|---|---|
| | | | Strongly Disagree | Disagree | Neither Agree Nor Disagree | Agree | Strongly Agree | Difference | *p*-Value |
| Panel A: Regulatory treatment | | | | | | | | | |
| Q6. The SEC's position is that virtual currency is not a security. Do you agree with this position? | 3.18 | 1.130 | 4.3 | 30.2 | 21.2 | 31.4 | 12.9 | 0.18 | 0.009 * |
| Q8. The IRS treats virtual currency as property, not currency. Do you agree with this position? | 3.36 | 1.062 | 3.5 | 22.0 | 22.0 | 40.4 | 12.2 | 0.36 | <0.001 * |
| Q4. In the U.S., cryptocurrency is not treated as tangible personal property. Do you agree with this statement? | 3.41 | 1.064 | 3.9 | 17.3 | 27.8 | 35.7 | 15.3 | 0.41 | <0.001 * |
| Q10. IFRS and the Australian Accounting Standards Board indicate that virtual currency meets the definition of an intangible asset. Do you agree with these positions? | 3.23 | 1.048 | 5.9 | 21.6 | 23.1 | 42.7 | 6.7 | 0.23 | 0.001 * |
| Q13. AICPA's Digital Asset Practice Aid indicates that virtual currency is not cash or cash equivalent. Do you agree with this position? | 3.43 | 1.201 | 5.9 | 22.0 | 15.3 | 36.9 | 20.0 | 0.43 | <0.001 * |
| Q15. The FASB, in their Board meeting on 11 May 2022, stated that virtual currency is not a financial instrument or financial asset. Do you agree with this position? | 2.72 | 1.125 | 12.5 | 38.4 | 18.8 | 25.1 | 5.1 | -0.28 | <0.001 * |
| Q5. Virtual currency is treated as a commodity by the Commodity Futures Trading Commission. Do you agree with this treatment? | 3.09 | 0.935 | 4.7 | 21.6 | 38.0 | 31.8 | 3.9 | 0.09 | 0.141 |
| Q14. If virtual currency becomes less volatile, it should be treated as a foreign currency. Do you agree with this statement? | 2.89 | 1.078 | 9.4 | 30.6 | 26.7 | 28.2 | 5.1 | -0.11 | 0.104 |

**Table 2.** *Cont.*

| Questions | Mean | Std. Deviation | Percentage Frequency | | | | | Test of Mean | |
|---|---|---|---|---|---|---|---|---|---|
| | | | Strongly Disagree | Disagree | Neither Agree Nor Disagree | Agree | Strongly Agree | Difference | *p*-Value |
| Q21. The SEC states that an Initial Coin Offering of a digital asset in order to raise capital should be treated as securities. Do you agree with this treatment? | 3.31 | 0.969 | 4.7 | 16.1 | 28.6 | 44.7 | 5.9 | 0.31 | <0.001 * |
| Q22. The international Swaps & Derivatives Association indicates that the current accounting for digital assets as intangibles does not reflect their economics and is therefore misleading. Do you agree with this statement? | 3.39 | 0.897 | 1.6 | 12.9 | 41.2 | 33.7 | 10.6 | 0.39 | <0.001 * |
| Panel B: Accounting treatment | | | | | | | | | |
| Q2. FASB must take a position on the accounting for virtual currency. | 4.23 | 0.825 | 2.0 | 1.6 | 8.6 | 47.5 | 40.4 | 1.23 | <0.001 * |
| Q11. IFRS indicates the virtual currency held as inventory should be treated at fair value less cost to sell. Do you agree with this treatment? | 3.60 | 0.959 | 2.4 | 12.5 | 22.4 | 48.6 | 14.1 | 0.60 | <0.001 * |
| Q12. Although virtual assets are held for sale, they are not tangible assets and do not meet the definition of inventory. Do you agree with this statement? | 3.59 | 1.003 | 2.0 | 14.9 | 22.7 | 43.1 | 17.3 | 0.59 | <0.001 * |
| Q16. Under Japanese GAAP, if virtual currency has an active market, the mark-to-market approach should be used. Do you agree with this approach? | 3.71 | 0.898 | 1.6 | 8.6 | 23.9 | 49.4 | 16.5 | 0.71 | <0.001 * |
| Q17. Under Japanese GAAP, if virtual currency is not on an active market, use cost. Do you agree with this approach? | 3.33 | 0.948 | 2.4 | 19.2 | 29.0 | 42.0 | 7.5 | 0.33 | <0.001 * |
| Q20. If virtual currency is impaired, impairment loss should be recognized. Do you agree with this treatment? | 3.75 | 0.921 | 1.6 | 10.2 | 17.6 | 52.5 | 18.0 | 0.75 | <0.001 * |

**Table 2.** *Cont.*

| Questions | Mean | Std. Deviation | Percentage Frequency | | | | | Test of Mean | |
|---|---|---|---|---|---|---|---|---|---|
| | | | Strongly Disagree | Disagree | Neither Agree Nor Disagree | Agree | Strongly Agree | Difference | *p*-Value |
| Q24. The impairment model is costly because a company is required to identify the lowest price and estimate fair value for their digital asset holdings throughout the entire holding period. Do you agree with this statement? | 3.18 | 0.934 | 5.1 | 16.9 | 36.9 | 37.3 | 3.9 | 0.18 | 0.002 * |
| Q19. If virtual currency recovers after a drop in value, do not mark-up. Do you agree with this treatment? | 2.61 | 1.088 | 12.5 | 42.4 | 22.4 | 16.9 | 5.9 | −0.39 | <0.001 * |
| Q23. Accounting for digital assets at cost less impairment does not reflect the economics of digital assets. Do you agree with this statement? | 3.38 | 1.001 | 2.7 | 17.6 | 30.2 | 37.3 | 12.2 | 0.38 | <0.001 * |
| Q18. Virtual currency should be treated as an indefinite-lived intangible asset subject to impairment. Do you agree with this treatment? | 2.89 | 1.139 | 11.8 | 29.0 | 24.7 | 27.8 | 6.7 | −0.11 | 0.111 |
| Q9. Bitcoins and other virtual currency transactions are traceable (e.g., there is an audit trail). | 3.13 | 1.155 | 9.0 | 23.9 | 22.0 | 35.3 | 9.8 | 0.13 | 0.074 |
| Panel C: Cryptocurrency impact | | | | | | | | | |
| Q1. Do you believe virtual currency in its current form (e.g., Bitcoin) are here to stay? | 3.75 | 0.936 | 2.7 | 8.2 | 18.4 | 52.9 | 17.6 | 0.75 | <0.001 * |
| Q3. The U.S. dollar will be dethroned internationally by the China's digital yuan. | 2.21 | 0.834 | 20.0 | 44.3 | 31.0 | 3.9 | 0.8 | −0.79 | <0.001 * |
| Q25. If the U.S. adopts a virtual currency (called CBDC) and paper money is eliminated, there will be less tax fraud and a significant increase in federal and state tax revenues. Do you agree with this statement? | 2.35 | 1.035 | 24.7 | 31.8 | 28.6 | 13.7 | 1.2 | −0.65 | <0.001 * |

**Table 2.** *Cont.*

| Questions | Mean | Std. Deviation | Percentage Frequency | | | | | Test of Mean | |
| --- | --- | --- | --- | --- | --- | --- | --- | --- | --- |
| | | | Strongly Disagree | Disagree | Neither Agree Nor Disagree | Agree | Strongly Agree | Difference | *p*-Value |
| Q7. If the U.S. adopts a virtual currency, many banks will disappear, and services will be reduced. Do you agree with this statement? | 2.45 | 0.998 | 14.1 | 46.7 | 23.1 | 12.5 | 3.5 | −0.55 | <0.001 * |
| Q26. Adoption of a governmental virtual currency will be an infringement on the privacy of your income and expenditures. | 3.11 | 1.210 | 8.6 | 25.9 | 28.2 | 20.8 | 16.5 | 0.11 | 0.162 |

This table presents the descriptive statistics for each survey question (N = 255). The responses to the 5-Likert scale are coded from 1 to 5 with 1 for "Strongly Disagree" up to 5 for "Strongly Agree". The "Mean" for each question can take values from 1 to 5. The percentage frequencies are in percentages and represent the proportions of each point of the 5-Likert scale. The test of Mean represents the test of the null hypothesis that the Mean of each question equals the neutral value of "3". The questions in each panel are listed in the same order as their discussion in the text. The numbering of the questions follows their order in the survey in which we randomized the questions in order to reduce order and recency biases. * = a *p*-value significance level of less than 1%.

### 6.3. Regulatory Treatment

The definition of virtual currency has received much attention from stakeholders and regulatory bodies. The results suggest that faculty generally agree with most of the proffered definitions. Accounting faculty agreed with the SEC that virtual currency is not a security (Q6). Likewise, they concurred with the IRS in treating virtual currency as property and not currency (Q8). Furthermore, accounting faculty agreed with the U.S. about not treating cryptocurrency as tangible personal property and with the IFRS and the Australian Accounting Standards Board that virtual currency meets the definition of an intangible asset (Q4, Q10). Additionally, there was an agreement with the AICPA's Digital Asset Practice Aid indicating that virtual currency is not cash or cash equivalent (Q13). Nevertheless, faculty had significant disagreement with the FASB's decision in their Board meeting on 11 May 2022, that virtual currency is not a financial instrument or financial asset (Q15). Similarly, there was no conclusive opinion on the virtual currency treatment as a commodity by the Commodity Futures Trading Commission (Q5), nor the treatment as a foreign currency when the virtual asset becomes less volatile (Q14). Overall, accounting faculty considered crypto currency as a form of intangible asset or property rather than as currency, cash, a security, or as a financial instrument.

While faculty defined digital currency as an intangible asset, they seemed to accept other forms of treatments within specific contexts. For example, they agreed with the SEC treating digital assets as securities for the purpose of raising capital in an initial coin offering (Q21). In the same context, faculty indicated that the current accounting for digital assets as intangibles is misleading since such classification does not reflect their economics—an opinion shared with the International Swaps & Derivatives Association (Q22), although in the two previous instances, faculty seemed to deviate from their mainstream definition of digital currency. This finding is evidence of the complexity of defining digital assets and the necessity for some flexibility to align with unique contexts.

### 6.4. Accounting Treatment

Next, the accounting treatment of digital currency was explored. The second question (Q2) had the highest mean, indicating a statistically strong agreement among faculty with the statement that FASB must take a position on the accounting for virtual currency. This result was congruent with previously discussed initial steps taken by FASB. More specifically, the accounting faculty agreed with the IFRS that virtual currency held as inventory should be treated at fair value less the cost to sell (Q11). However, faculty indicated that virtual assets held for sale are not tangible assets and therefore do not meet the definition of inventory (Q12). Furthermore, faculty significantly agreed with Japanese GAAP in accounting for virtual currency using the mark-to-market approach when traded in an active market, and alternatively, the cost method in the absence of an active market (Q16, Q17).

Turning to virtual currency impairment, accounting faculty agreed with recognizing impairment loss (Q20) despite their belief that the impairment model is costly (Q24). They also objected to the idea of not marking up virtual currency when it recovers after a drop in value (Q19). The mark-up of virtual currency when it recovers is consistent with their belief that accounting for digital assets at cost less impairment does not reflect the economics of digital assets (Q23).

There was no consensus among faculty as to treating virtual currency as an indefinite-lived intangible asset subject to impairment (Q18). With no statistically significant deviation from the neutral opinion, this treatment remains unsettled. There was also weak agreement among faculty (significance at 7.4%) that Bitcoins and other virtual currency transactions are traceable with an available audit trail (Q9).

### 6.5. Cryptocurrency Impact

A few survey questions assessed faculty's general perception of the anticipated impact of digital currency and its future development. There was a strong belief that virtual

currency in its current form is here to stay (Q1). However, faculty statistically disagreed with the prediction that the U.S. dollar will be dethroned internationally by the China's digital yuan (Q3). There was also a significant disagreement with the idea that the U.S. adoption of a virtual currency (called CBDC) and the elimination of paper money will result in less tax fraud and a significant increase in federal and state tax revenues (Q25), or such adoption will eliminate few banks and reduce financial services (Q7). Nonetheless, faculty remained divided on the prediction that the adoption of a governmental virtual currency will be an infringement on the privacy of someone's income and expenditures (Q26).

## 7. Discussion of Results

As indicated in Section 6.1 and in Table 1, the survey respondents were predominantly aged 50–59, tenured, at the rank levels of associate or professor, experienced, and PhDs. The present authors suggest that the study's demographics provide a sense of the level of expertise of the respondents, which is important in considering the weight of their opinions.

Accounting faculty generally diverged regarding digital currency definitions of a currency, cash or cash equivalents, a security, a financial instrument, or financial asset. They tended to consider digital currency as a form of intangible asset. Faculty accepted other digital currency definitions within specific contexts. For instance, faculty concurred with the SEC treatment of digital currency as a security when raising capital in an initial coin offering. Furthermore, faculty indicated, in the context of derivatives, that classifying digital assets as intangibles is misleading and not reflective of their economies. The multitude of definitions and classifications given to digital currencies is evidence of their complexity.

Congruent with Japanese GAAP, faculty favored the mark-to-market when the cryptocurrency is traded on an active market, and the cost method otherwise. Overall, faculty recommended the use of the cost method for virtual currency with impairment loss recognition. Furthermore, they considered that, due to the economic nature of cryptos, mark-up is permissible when recovery occurs. However, they recognized that the impairment model is costly to implement.

In addition, faculty do not expect a major disruption in the post-adoption period of digital currencies. Globally, they still foresee the continued dominance of the U.S. dollar. They predict no impact of digital currencies on the economic role of financial institutions and no impact on the level of tax fraud and tax revenues.

## 8. Limitations

The results of this exploratory research are not generalizable to the population of accounting faculty. Most respondents were from the U.S., the sample was not randomly generated, and there was a relatively low response rate. However, due to the novel nature and the fast developments within the digital currency field, it was assumed that faculty with limited knowledge of the topic refrained from responding to the survey. Thus, we are confident that the respondents were knowledgeable enough to formulate an opinion. That the respondents were informed about the field of digital currencies was somewhat reflected in the additional in-depth comments and feedback provided in response to the open-ended question, which are compiled in Appendix A.

## 9. Conclusions

Faculty answers to the survey's 26 virtual currency questions fulfilled the purpose of the study, which was to provide insights into accounting faculty perceptions of how crypto assets should be defined and reported. The development of cryptocurrencies has impacted the pillars of regulatory and centralized monetary policy, but the Financial Accounting Standards Board (FASB) has been slow in developing a position on how to account for cryptocurrencies. The FASB only recently adopted a new standard, which will be effective towards the end of 2024. An auxiliary purpose of this study was to show the path the FASB has taken in developing accounting standards for crypto assets, to outline positions other authorities and agencies have taken, and to discuss Central Bank Digital Currencies. Since

the United States and other countries are considering replacing their fiat currency with a digital currency, central banks will play increasingly important roles.

**Author Contributions:** Conceptualization, methodology, software, validation, formal analysis, investigation, resources, data curation, writing—original draft preparation, writing—review and editing, visualization, supervision, project administration, funding acquisition, D.L.C., D.L.A. and A.K. contributed equally through every process of the research study. All authors have read and agreed to the published version of the manuscript.

**Funding:** This research was funded by the Coles College of Business Research and Development Committee at Kennesaw State University under the grant number Spring23-03.

**Data Availability Statement:** Data is unavailable due to proprietary and privacy reasons.

**Conflicts of Interest:** The authors declare no conflict of interest.

**Appendix A**

Responses to the open question: Please indicate below anything about the treatment of virtual currency that was not addressed in the above questions and statements.

| |
|---|
| I view virtual currency (in its current format) as any other foreign currency. |
| I think that regulators should start implementing rules and standards that govern digital currency-based transactions. Rules should close any gaps related to CBDC-related taxes, and CBDC transfer from one bank/country to another. Will money laundering rules be applied to CBDC as well? Will CBDC be used as a new way to commit fraud schemes which are hard to detect? Will the SEC rules change/adapt to be able to prosecute fraudsters using CBDC? How would they measure financial loss and injury if the market value of the digital currency keeps fluctuating? There are many issues that need to be discussed and hashed out to ensure the safe use of the CBDC. In other words, we need more enforceable regulations. Let us remember that 2008 financial crash happened partially due to the absence of regulations governing the use of derivatives and debt swaps by financial services organizations (banks and mortgage lenders). |
| It is a type of foreign currency or investment (depending on use) |
| The competing issues of privacy and traceability, the impact of virtual currency on compliance with AML (anti-money laundering) and know your customer banking regulations. |
| The environmental impacts of various virtual currencies, especially given expanded ESG reporting requirements. |
| FASB is the standard setting body for future consideration of virtual currencies. |
| We need to be more aggressive in reflected digital currency in the US. |
| The appropriate valuation is LCM if it's not considered a currency. But as it is increasingly used as a currency substitute, I think it should be considered a currency, regulated, and valued at market. The constant swings in value make it unlikely this will happen any time soon. |
| I agree that there should be a consistent definition—leaving the issue undecided is a mistake. However, I do not believe that there is a consensus on what precise characteristics are relevant. I feel much more research is warranted, BUT in the meantime, there should be an easy to understand and apply accounting standard in place. I strongly disagree that such virtual currency is a security—that approach neglects the need to further understand the nature of and the various differences in this novel financial instrument. |
| The underlying economics argument is invalid. If the digital currency collapses, then its holders have nothing at all to show for it and have no real recourse. In short, they lose everything. |
| The experts on financial programs seem to indicate that there are different types of cryptocurrencies, e.g., Bitcoin versus Ethereum versus Litecoin. Will accounting regulations provide sufficient transparency so that these differences are apparent to the investors? It seems most people do not understand cryptocurrency and cannot make informed investment decisions about it. When Warren Buffet and Charlie Munger disparage crypto currencies, it makes one wonder if they are suspect places to invest money. |
| I wish you'd just ask how I think it should be treated. I think it should be treated as foreign currency/cash equivalents as long as there is an active market and/or it can be used to purchase goods and services (e.g., Bitcoin and Ethereum, the proposed digital dollar). For inactive markets or digital currencies that are not widely accepted, I would treat them as any other financial investment under GAAP (i.e., level 1, 2, and 3 input categories to fair value). However, if I were king of accounting, I'd also change any level 2 or 3 financial investment to cost, with gains and losses recognized at the time of a sale, so ultimately, I'd love to see inactive virtual currencies accounted for at cost with FN disclosure around estimated fair value. |
| Virtual/digital currencies should be treated the same as sovereign foreign currency. These should be valued at the market rate, and the value should be regularly updated in the accounts. However, virtual currency holdings that have been "staked" are a financial security and will need different accounting treatment. |

| |
|---|
| I honestly believe that the way that virtual currency is being used currently is the equivalent of a security being traded on an exchange (Coinbase or similar exchange). Therefore, it should be accounted for in the same way that securities are accounted for, adjusted to fair value when preparing the financial statements. Only if there is a change in the nature of virtual currency in the future should it be treated in some other way. |
| I believe that treating virtual currencies as an investment is the best model. They should be valued using mark-to-market (FMV), similar to an investment. The fact they can be used as a currency is merely a convenience—you could sell a trading security and take the cash to make a purchase. Eliminating the middle step is merely a convenience. If a government controls the virtual currency, the FMV of the virtual currency could be treated as a foreign currency and translated at an established exchange rate. |
| The most basic question is whether the government should make private digital currency illegal or questions about more oversight. Should the US SEC and other SEC's require digital currencies to register and produce 10-k's? Should U.S. Congress disallow digital currency. Digital currency is used by criminals to circumvent more expensive means of payment and money laundering. I am bewildered still as to why the government permits it.<br>RE: Central Bank Digital Currency. In my opinion, CBDC should not be treated like a security but as cash because central bank currency serves as a numeraire and central bank policy makers' have established policies, legal guidelines, and resources to stabilize their currencies (things that private digital currency issuers do not). So, perhaps a separate question about the treatment of CBDC. |
| There are feasible circumstances where my answers could be different depending upon some attributes or situations not discussed in the question. The best accounting is to treat virtual currency as an investment and that changes in value either be treated as per current impairment accounting or similar to gains and losses due to foreign currency translations. |
| Much of my views on this topic have some 'dependence' on time frame. VC is here to stay but is going to take significant time to become mainstream. The question, for example, about VC overtaking standard banking—my opinion is that it will not do so in the near term but will eventually become more common. This is similar to so many other things we do, from ordering groceries online to banking online to paying with our phones, etc. There are any things we could not imagine people "trusting" enough to use them, and now they are standard. |
| The volatility of digital currency and the lack of controls over its creation are overly concerning to me. Definitely still the wild west and I wonder what would be the determining factor(s) that would bring digital currency more legitimacy, less risk and fast adoption by the public. I do not trust its permanency and so I have no investment exposure to it at all. |
| The focus of the question is what "should be" happening from an accounting viewpoint, only. Try an accounting approach based on what is actually happening. It is my understanding that virtual currencies are not being used as currency to fund transactions. Instead, virtual currencies are being used for quick trading gains/losses.<br>I view these currencies as non-fungible tokens. |
| Digital currency, if we can call it that, is not monolithic. Bitcoin and Ethereum function differently, so are many other chains or coins that aim to satisfy unmet demand for exchange and record keeping not offered by fiat and banks. My response large reflect my opinion on Bitcoin but not other more centralized but more liquid and functional crypto. |
| I would think that digital currencies should be valued at the lower of cost or net realizable value (NRV being proceeds minus cost to sell or dispose of), assuming the digital currency has an active market. Any digital currency that does not have an active market is suspect, and I'm not sure how such digital currencies should be treated. . .perhaps as intangible assets that need to be reviewed for impairment at each balance sheet date.<br>My answers might differ depending on whether the digital currency is controlled by the U.S. government vs. the government of China. I do not particularly trust the Chinese government. |
| Distinguishing amongst virtual currencies rather than treating them all uniformly |
| The question of SEC involvement in regulation is important. The Ethereum Foundation which controls Ethereum is a Swiss based non-profit. The SEC has never tried to regulate non-profits and this shift is concerning. |
| Crypto currency does not fit any of the current definitions of cash, currency, cash equivalent, securities, inventory, or financial asset. I propose creating a new category "digital asset" and using fair value accounting for this asset. |
| Any currency that is unable to "explain its existence" in a manner that is simple, succinct, and straightforward causes a "Ponzi Scheme" alert to surface for me. |

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
