# Peer review of "How Should Cryptocurrencies Be Defined and Reported? An Exploratory Study of Accounting Professor Opinions"

_jrfm, doi:10.3390/jrfm17010003_

Round 1

Reviewer 1 Report

Comments and Suggestions for Authors

1. The introduction should be expanded to clearly identify the purpose of the study and highlight the problem. The study covers three dimensions (regulatory treatment, accounting treatment, and cryptocurrency impact), but it is not clear from the introduction what specific problem the authors are addressing, why they are addressing it through these dimensions, and why it is needed to survey the accounting professors to solve the problem. Perhaps the research problem would be more clearly identified if the authors were to present more fully what research has already been done in this field, and then it might become clear that the view of accounting professors have not been heard so far. And it view could possibly make a significant contribution to better accounting and regulation of cryptocurrencies.

2. Section 4.2 does not indicate the final number of respondents. It is up to the reader to calculate this number on the basis of Table 1. It is recommended that this number (255) is given in the text.

3. The presentation of the results of the study is rather fragmented, the information is not properly structured and the information is inconsistently presented. There is also a serious lack of figures that would allow a clear and quick overview of the results. Table 2 in the paper is too long and should be broken down according to some logic. Perhaps this could be broken down into the three groups of questions (the three dimensions of the study) identified by the authors. At present, when reading the text of sections 5.3; 5.4 and 5.5, the reader is constantly referred back to Table 2, where the questions are listed according to their numbers, but not in the same way as they are grouped according to the logic of the question groups. This results in a constant "distraction" between questions and answers. Thus, as mentioned above, one possible option is to group the questions in Table 2 along the three dimensions and present the answers in the form of tables or figures in each of the subsections (Regulatory Treatment; Accounting Treatment, Cryptocurrency Impact). It is also recommended that each of the dimensions be expanded with the answers from Annex 1. At present, the open-ended responses in Annex 1 are not grouped, so it is not clear which dimension of the study they relate to and how these responses contributed to the research problem. It should also be noted that the authors have extensively described the demographics of the respondents (Table 1), but these data have not been highlighted in any way in the analysis of the respondents' answers. I.e. nowhere was it noted that, for example, respondents who were older or more experienced, or who were distinguished by some other attribute, had a different (unique) assessment of one or another issue. In the absence of any presentation of responses by demographic breakdown, these demographic data are of little value for the study.

4. As the introduction does not explicitly identify the problem, the conclusions do not, accordingly, indicate how the problem has been solved, i.e. how the results have contributed, or could contribute, to a fairer and clearer process for the management and accounting of cryptocurrencies.

Author Response

Thank you for your constructive input. We have added updated developments, which we have highlighted in blue. We have made a few edits that are highlighted in yellow. We have responded to each of your comments with new text highlighted in yellow. We hope that you find our responses and changes acceptable. We greatly appreciate the time and effort you put into reading and commenting on our paper. 

Reviewer 2 Report

Comments and Suggestions for Authors

“How should Cryptocurrencies Be Defined and Reported? An Exploratory Study of Accounting Professor Opinions” presents the results of a survey of 255 accounting professors on their opinions regarding the treatment of cryptocurrency. Although the response rate may be low (3%), obtaining 255 responses from accounting professors is impressive. The size alone makes this a valuable contribution to the existing literature.

The article itself is rather concise as it primarily provides the survey results. This, however, is appropriate as the position of accounting experts has implications for a range of studies regarding cryptocurrencies. It provides a good overall introduction with an overview of the guidelines of the Digital Asset Working Group.  However, the paper is lightly cited.  In a paper on defining cryptocurrency, the previous literature on categorization of cryptocurrency should be addressed even if only to make the readers aware of the existence of other perspectives.  Applicable literature includes:

Ankenbrand, T.; Bieri, D.; Cortivo, R.; Hoehener, J.; Hardjono, T. (Eds.) Proposal for a comprehensive (crypto) asset taxonomy. In Proceedings of the 2020 Crypto Valley Conference on Blockchain Technology (CVCBT), Rotkreuz, Switzerland, 11–20 June 2020; IEEE: Piscataway, NJ, USA, 2020. 

Fokri, W.N.I.W.M. Classification of cryptocurrency: A review of the literature. Turk. J. Comput. Math. Educ. TURCOMAT 2021, 12, 1353–1360

Di Matteo, G.; Za, S.; Ulrich, K. Initial Coin Offering: A Taxonomy Based Approach to Explore the Field. In Proceedings of the MENACIS2021, Agadir, Morocco, 11–14 November 2021.

Watters, C. Digital Gold or Digital Security? Unravelling the Legal Fabric of Decentralised Digital Assets. Commodities 20232, 355-366. https://doi.org/10.3390/commodities2040020

While scammers may seek payment in crypto, money laundering in not particularly common (as claimed at ln 44) with the chainalysis report on crypto crime placing illicit crypto transactions significantly lower than UN estimates of money laundering with traditional currencies. If making such claims, it’s important to at least address evidence to the contrary. See https://go.chainalysis.com/2023-crypto-crime-report.html

After addressing these minor issues, this will be an important contribution for those studying crypto regulation.

Author Response

We appreciate your constructive input and have responded to each of your comments. Thank you for helping us improve our work. We especially think the paper was significantly improved by adding the Literature review based on your suggested references. We appreciate your support in moving this paper towards publication. Please note that we have updated information on the FASB. These minor changes are highlighted in blue. Text added based on your input is highlighted in yellow.
